# Concurrent Validity of Virtual Reality-Based Assessment of Executive Function: A Systematic Review and Meta-Analysis

**DOI:** 10.3390/jintelligence12110108

**Published:** 2024-10-31

**Authors:** Si-An Lee, Ji-Yea Kim, Jin-Hyuck Park

**Affiliations:** 1Department of ICT Convergence, The Graduate School, Soonchunhyang University, Asan 31538, Republic of Korea; sianison3@sch.ac.kr (S.-A.L.); kimkim2@sch.ac.kr (J.-Y.K.); 2Department of Occupational Therapy, College of Medical Science, Soonchunhyang University, Asan 31538, Republic of Korea

**Keywords:** virtual reality, executive function, neuropsychological assessment, ecological validity

## Abstract

This meta-analysis investigated the concurrent validity between virtual reality (VR)-based assessments and traditional neuropsychological assessments of executive function, with a focus on subcomponents such as cognitive flexibility, attention, and inhibition. A total of 1605 articles were identified through searches of PubMed, Web of Science, and ScienceDirect from 2013 to 2023. After removing duplicates, 1313 articles were screened based on their titles and abstracts, with 77 articles selected for full-text eligibility review. Of these, nine articles fully met the inclusion and exclusion criteria for this study. The effect size for overall executive function was assessed, with subcomponents categorized based on the specific assessment tools used in the studies. The effect size for each subcomponent—cognitive flexibility, attention, and inhibition—was then analyzed to provide a more detailed understanding of their relationships with traditional measures. The results revealed statistically significant correlations between VR-based assessments and traditional measures across all subcomponents. Additionally, sensitivity analyses confirmed the robustness of the findings, even when low-quality studies were excluded. These results support the use of VR-based assessments as a valid alternative to traditional methods for evaluating executive function. Moreover, the study highlights the potential of VR-based assessments as a valid alternative to traditional methods, emphasizing the need to address variability in executive function subcomponents and integrate diverse cognitive and motor metrics for greater ecological validity.

## 1. Introduction

Recently, executive functions have increasingly been defined as separable, yet interrelated components are involved in goal-directed thinking and behavior ([7]). Many studies have identified three key subcomponents of executive function: working memory, inhibition, and cognitive flexibility ([6]; [20]). However, [20]’s ([20]) model was not intended to be comprehensive, acknowledging the possibility of additional separable subcomponents. Recent studies have further reinforced the importance of viewing executive functions as distinct subcomponents rather than as a single unified function. Neurological evidence from large-scale brain networks supports this perspective, showing that while these components are anatomically and functionally distinct, they interact within integrated circuits to support complex cognitive tasks ([9]; [34]). This approach provides a more refined framework for both theoretical models and practical assessments of executive function.

Executive functions, considered higher-order cognitive abilities, are essential for managing tasks across various age groups and clinical populations. Impairments in executive functions can undermine academic performance and reduce the ability to carry out independent activities in daily life ([4]; [17]). Furthermore, impairments in executive functions also negatively affect disease management, as individuals may fail to recognize the severity of their health problems ([14]). Therefore, assessing and monitoring executive functions is vital in predicting whether individuals can effectively manage everyday tasks and activities related to education and daily life.

Executive functions have been traditionally assessed through paper-and-pencil neuropsychological assessments. These assessments involve a wide range of performance-based measures ([22]), including tasks such as the Trail Making Test (TMT) ([26]), the Stroop Color-Word Test (SCWT) ([12]), and the Wisconsin Card Sorting Test (WCST) ([13]). Additionally, assessments conducted through questionnaires, such as the Behavior Rating Inventory of Executive Function (BRIEF) ([10]), are also available. Given the multifaceted nature of executive functions, comprehensive evaluations often utilize neuropsychological batteries such as the Delis–Kaplan Executive Function System (D-KEFS) ([5]) and the Cambridge Neuropsychological Test Automated Battery (CANTAB) ([3]).

However, traditional paper-and-pencil methods for neuropsychological assessment lack similarity to real-world tasks and fail to adequately simulate the complexity of everyday activities ([30]), resulting in low ecological validity and limited generalizability. To overcome these limitations, recent efforts have focused on developing neuropsychological assessments using virtual reality (VR) technology. VR allows subjects to engage in real-world activities implemented in virtual environments ([23]). With advancements in technology, VR enables natural movement recognition, facilitating immersion in virtual environments ([8]). Therefore, neuropsychological assessments evaluated through activities provided via VR offer ecological validity. Additionally, VR provides a controlled environment, ensuring safety and allowing for the objective and automatic measurement and management of responses to activities ([31]).

A previous meta-analytical study has revealed significant moderate correlations between VR-based assessments and traditional or computerized neuropsychological assessments. However, despite the multifaceted nature of executive functions, previous research has considered executive functions as a single cognitive function, aggregating attention, impulsivity, cognitive and motor inhibition, and executive function itself, which represents a limitation ([21]). This study offers a more granular examination of the subcomponents of executive functions, which adds depth to the analysis. The breakdown of subcomponents such as inhibition, working memory, and cognitive flexibility allows a more detailed validation of VR-based assessment, rather than treating executive functions as a monolithic construct.

Therefore, this study conducted a systematic review and meta-analysis to assess the concurrent validity of VR-based assessments of executive functions in comparison to traditional neuropsychological assessments. Concurrent validity refers to the extent to which a new test correlates with an established one when both are administered simultaneously ([11]). Our analysis focuses on whether VR-based assessments yield comparable results to traditional methods in evaluating executive function. Additionally, we performed a detailed analysis of the subcomponents of executive functions—such as inhibition, working memory, and cognitive flexibility—to determine whether VR-based assessments reliably measure these cognitive abilities in line with traditional assessments.

## 2. Methods

This systematic review and meta-analysis was conducted in accordance with the Preferred Reporting Items for Systematic Review and Meta-Analyses (PRISMA) guidelines. 

### 2.1. Literature Search

In February 2024, we conducted a search of published literature in the following 3 databases: PubMed, Web of Science, and ScienceDirect. The following keywords were used: “Virtual Reality” AND “Executive function*”. The search period covered from 2013 to 2023.

### 2.2. Study Eligibility

The selected studies met the following inclusion criteria: (1) they used assessments to evaluate executive function, (2) they included assessments that utilized VR, (3) they were published in English, and (4) they were full-text articles. Studies that did not evaluate executive function, as well as review articles and case studies, were excluded. Additionally, studies were excluded if they did not provide sufficient data to identify the correlation coefficients between VR-based assessments and traditional paper-and-pencil tests.

### 2.3. Study Screening

Two reviewers independently screened the titles and abstracts of the studies, followed by a full-text review to identify eligible articles. In cases where discrepancies arose during the screening process, the two reviewers discussed the differences. If a consensus could not be reached, a third reviewer was consulted to resolve the disagreement.

### 2.4. Data Extraction

Two independent reviewers extracted data from the studies that met the inclusion criteria. The following variables were extracted: (1) first author and publication year, (2) program name, (3) mean age of participants, (4) proportion of male participants, (5) number of participants, (6) clinical status of the sample, (7) characteristics of VR-based assessments, (8) outcome measure, and (9) statistical data (Pearson’s r value).

### 2.5. Assessment of Study Quality

Two reviewers independently assessed the quality of the studies using the QUADAS-2 (Quality Assessment of Diagnostic Accuracy Studies 2) checklist ([33]). In cases where discrepancies arose, the two reviewers discussed the differences. If they were unable to reach a consensus, a third reviewer was consulted to resolve the disagreement.

### 2.6. Data Analysis

Comprehensive Meta-Analysis Software (CMA) Version 3 was employed to examine the association between the VR-based assessment of executive function and traditional paper-and-pencil executive function assessments. To calculate interrater reliability, Cohen’s Kappa analysis was conducted. To accurately account for sample size in each study, Pearson’s r values were transformed into Fisher’s z, and all analyses were conducted using Fisher’s z values. The heterogeneity of the studies was evaluated using I^2^. When heterogeneity was high (I^2^ > 50%), a random-effects model was applied; a fixed-effects model was used when heterogeneity was low. To assess the robustness of the results, sensitivity analyses were conducted by excluding lower-quality studies. Additionally, subgroup analyses were performed to explore potential moderators and examine whether specific factors, such as VR tool type or population characteristics, influenced the association between VR-based and traditional assessments. Publication bias was assessed using funnel plots and Egger’s regression test.

## 3. Results

### 3.1. Interater Agreement 

The interrater reliability between the two reviewers for study screening, data extraction, and quality assessment using Cohen’s Kappa was 0.89.

### 3.2. Study Selection

Figure 1 presents the PRISMA flow chart for the study selection process. The literature searches in the PubMed, Web of Science, and ScienceDirect databases identified a total of 1605 articles, of which 292 were removed due to duplication. Subsequently, after screening titles and abstracts, 1236 articles were excluded. Finally, nine articles meeting the inclusion criteria were selected for the study.

### 3.3. Study Characteristics

Table 1 summarizes the characteristics of the nine included studies. The studies included participants across a wide range of ages, from children to older adults. Two of the studies focused on children, while the remaining seven studies involved adult participants. The clinical status of the sample varied, with four studies including healthy and clinical participants, four studies including only healthy participants, and one study focusing solely on clinical participants. The clinical conditions represented in the sample included mood disorders, psychosis spectrum disorders, attention-deficit/hyperactivity disorder, Parkinson’s disease, and cancer. A total of eight VR-based assessments were used. Four assessments were adaptations of existing assessments (e.g., TMT-A, TMT-B) adapted for VR, while the other four assessed executive functions within virtual environment scenarios (e.g., kitchen, home). All outcome measures were administered using paper-and-pencil tests.

### 3.4. Study Quality

Figure 2 shows the results of the quality assessment of the studies using QUADAS-2. The risk of bias in patient selection was considered high in three studies, as all of them employed a case–control design ([2]; [28]; [32]). Additionally, two studies did not enroll patients using a consecutive or random sampling method ([28]; [32]). The risk of bias in patient selection was classified as unclear in one study ([16]) because it was not clear whether patients were assigned consecutively or randomly, and exclusion criteria were not specified, making it uncertain whether inappropriate exclusions were avoided. Thus, a total of four studies were classified as low-quality in terms of patient selection. 

The risk of bias in the index test was rated high across all studies because none of the VR-based tools included predefined thresholds for evaluation. While the absence of these predefined thresholds contributed to the high-risk rating in the index test, it was considered non-critical for this study as it did not affect the main results of the meta-analysis. For the flow and timing and reference standard, most studies adhered to acceptable practices. Despite these limitations, all studies were initially included for the analysis of overall executive function. A sensitivity analysis was then conducted by excluding low-quality studies to assess the robustness of the overall results. For the analysis of executive function subcomponents, all available studies, including the low-quality ones, were used to ensure sufficient statistical power.

### 3.5. Results of the Meta-Analysis

In this study, executive function was subdivided into the following subcomponents: attention, inhibition, and cognitive flexibility. The outcome measures for attention included the TMT-A, Color Trails Test-1, and attentive matrices. The outcome measures for inhibition included the Stroop test (ST)-errors and ST-time, and the D-KEFS Color-Word Interference Test (Inhibition, Inhibition/Switching). The outcome measures for cognitive flexibility included the TMT-B, TMT-BA, and Color Trails Test-2 (Table 2).

#### 3.5.1. Overall Executive Function

A total of nine studies were used to analyze overall executive function. The results indicated a statistically significant correlation between the VR-based assessment and traditional assessment of executive function, and the pooled Fisher’s z value was 0.454 (95% CI: 0.285–0.623; I^2^ = 75.05, Figure 3). Due to the high heterogeneity, a random-effects model was adopted. Figure 4 shows the funnel plot of the overall executive function. Funnel plots and Egger’s test indicated significant evidence of publication bias in the nine studies (*p* < 0.05).

#### 3.5.2. Sensitivity Analysis

After excluding four low-quality studies (with high or unclear risk of bias in patient selection), the results indicated a statistically significant correlation between the VR-based assessment and traditional assessment of executive function. The pooled Fisher’s z value was 0.485 (95% CI: 0.219–0.751; I^2^ = 79.959, Figure 5). Due to the high heterogeneity, a random-effects model was adopted. Egger’s test indicated no significant evidence of publication bias in the remaining five studies (*p* > 0.05).

#### 3.5.3. Attention

A total of five studies were used to analyze attention. The results indicated a statistically significant correlation between VR-based assessment and traditional assessment in terms of attention, and the pooled Fisher’s z value was 0.618 (95% CI: 0.204–1.032; I^2^ = 89.793, Figure 6). Due to the high heterogeneity, a random-effects model was adopted. Egger’s test indicated no significant evidence of publication bias in the five studies (*p* > 0.05).

#### 3.5.4. Inhibition

A total of two studies were used to analyze inhibition. The results indicated a statistically significant correlation between VR-based assessment and traditional assessment in terms of inhibition, and the pooled Fisher’s z value was 0.282 (95% CI: 0.09–0.483; I^2^ = 18.02, Figure 7). Due to the low heterogeneity, a fixed-effects model was adopted. Publication bias was not analyzed because the number of studies was too small.

#### 3.5.5. Cognitive Flexibility

A total of five studies were used to analyze cognitive flexibility. The results indicated a statistically significant correlation between VR-based assessment and traditional assessment in terms of cognitive flexibility, and the pooled Fisher’s z value was 0.498 (95% CI: 0.255–0.741; I^2^ = 73.069, Figure 8). Due to the high heterogeneity, a random-effects model was adopted. Egger’s test indicated no significant evidence of publication bias in the five studies (*p* > 0.05).

### 3.6. Subgroup Analysis

#### 3.6.1. VR-Based Assessment Properties

We conducted a subgroup analysis to determine whether the properties of VR-based assessments moderated the correlation between VR-based assessments and traditional assessments. The analysis was performed on overall executive function, as well as the subcomponents of attention and cognitive flexibility. Nine studies were included in the overall executive function analysis, while five studies each were included for both the attention and cognitive flexibility analyses. The results indicated that VR-based assessment properties did not moderate the relationship between VR-based and traditional assessments in the overall executive function analysis (Q-value(df) = 0.364(1), *p* = .546). Similarly, no moderating effect of VR properties was found in the attention analysis (Q-value(df) = 0.307(1), *p* = .579) or the cognitive flexibility analysis (Q-value(df) = 1.508(1), *p* = .219).

#### 3.6.2. Clinical Status of the Sample

We conducted a subgroup analysis to determine whether the clinical status of the sample moderated the correlation between VR-based assessments and traditional assessments. The analysis was performed for overall executive function, as well as the subcomponents of attention and cognitive flexibility. The results indicated that clinical status did not moderate the relationship between VR-based and traditional assessments in the overall executive function analysis (Q-value(df) = 4.18(2), *p* = .124) or the cognitive flexibility analysis (Q-value(df) = 5.671(2), *p* = .059). However, a moderating effect of clinical status was found in the attention analysis (Q-value(df) = 17.215(1), *p* < .001).

#### 3.6.3. Age of the Sample

We conducted a subgroup analysis to determine whether the age of the sample moderated the correlation between VR-based assessments and traditional assessments. This analysis was performed for overall executive function only. The results indicated that age did not moderate the relationship between VR-based and traditional assessments in the overall executive function analysis (Q-value(df) = 0.305(1), *p* = .581).

## 4. Discussion

This study investigated the concurrent validity of VR-based assessments of executive function in comparison to traditional neuropsychological assessments. Nine studies were included, examining overall executive function as well as the subcomponents of cognitive flexibility, attention, and inhibition. The results demonstrated significant correlations between VR-based assessments and traditional measures, supporting VR as a valid method for evaluating these cognitive functions. Although high heterogeneity was observed, indicating variability in the study outcomes, sensitivity analyses confirmed the robustness of the findings, even when lower-quality studies were excluded. These findings suggest that VR-based assessments are a reliable alternative to traditional methods for evaluating overall executive function and its subcomponents.

The quality assessment using QUADAS-2 revealed that four studies had either a high or unclear risk of bias in the patient selection domain. To assess the impact of study quality, a sensitivity analysis was conducted by excluding these four low-quality studies. Despite their exclusion, the results remained statistically significant. Although high heterogeneity persisted, publication bias was no longer detected. This suggests that the significant correlations observed are likely to reflect genuine results, rather than being influenced by biased or poorly conducted studies ([18]). In the index test domain, all studies were assessed as having a high risk of bias due to the absence of predefined thresholds for evaluation. However, no sensitivity analysis was conducted for the index test domain because the comparisons between VR-based assessments and traditional paper-and-pencil tests were based on correlations, not on predefined thresholds. As a result, the absence of thresholds did not affect the main results, and a sensitivity analysis for this domain was deemed unnecessary.

The subcomponents of inhibition, attention, and cognitive flexibility were analyzed separately to gain a more detailed understanding of how VR-based assessments relate to traditional neuropsychological tools. These subcomponents were chosen because at least two of the studies that were assessed each used similar paper-and-pencil tests, allowing for meaningful comparisons across studies. The analysis revealed significant correlations between VR-based and traditional assessments for all three subcomponents, supporting the concurrent validity of VR-based assessments in measuring distinct aspects of executive function. Our findings are consistent with previous studies that demonstrated correlations between VR-based assessments and traditional neuropsychological evaluations ([21]). Additionally, the results of the nine studies considered in this paper align with our findings, as most reported significant correlations. However, while prior studies treated executive function as a single construct without providing an in-depth analysis of its subcomponents ([1]; [21]), our research offers independent analyses of key subcomponents, such as cognitive flexibility, attention, and inhibition. This distinction is important, as multiple studies have indicated that executive function consists of separable but interrelated components ([7]; [9]). Therefore, our study’s focus on analyzing these subcomponents independently adds meaningful insights. Nevertheless, we observed high heterogeneity, particularly for attention and cognitive flexibility, indicating variability across studies. Despite this variability, the significant correlations suggest that VR-based assessments can still accurately evaluate key aspects of executive function.

This study observed high heterogeneity across the included studies. Subgroup analyses were conducted to explore potential sources of heterogeneity, examining moderators such as VR assessment properties, clinical status, and age. The results indicated that VR assessment properties—whether they were direct adaptations of traditional paper-and-pencil tests or assessments conducted within virtual scenarios—did not significantly moderate the relationship between VR-based assessments and traditional assessments for overall executive function, attention, or cognitive flexibility. In contrast, clinical status was a significant moderator only in the attention analysis, where the combined and clinical populations exhibited different correlation patterns compared to the healthy population. However, clinical status did not significantly moderate the results for overall executive function or cognitive flexibility. No significant moderating effect was found for age in the overall executive function analysis. These subgroup analyses provided insights into potential sources of heterogeneity, but the remaining high variability underscores the complexity of comparing VR-based and traditional assessments across diverse contexts. Future studies should investigate additional factors, such as task difficulty and environmental context, to better understand and reduce heterogeneity.

Publication bias was assessed using funnel plots and Egger’s regression test. However, a sensitivity analysis was performed excluding studies classified as low quality by QUADAS-2 assessment, and publication bias was no longer detected, strengthening the validity of the results. This suggests that the initial presence of bias was largely driven by lower-quality studies, which may have inflated the overall correlations. While this mitigates concerns about publication bias, the small number of studies included in the meta-analysis may still limit the ability to fully assess such bias. Future research should aim to increase sample size and ensure the comprehensive reporting of both significant and non-significant results to provide a more balanced and accurate picture of the effectiveness of VR-based assessments.

This study confirms that VR-based assessments of executive function are valid and demonstrate considerable ecological validity compared to traditional paper-and-pencil tests. Previous studies have shown that some VR-based assessments were direct adaptations of paper-and-pencil tests, while others employed entirely new virtual scenarios to assess executive function. In cases where VR-based assessments were a direct adaptation, the use of VR equipment did not significantly affect the assessment outcomes, resulting in strong correlations with traditional neuropsychological evaluations. This suggests that once individuals adapt to the VR system, the new medium does not substantially influence cognitive function assessments ([24]; [32]). In fact, VR assessments offer immersive, interactive environments that simulate real-world tasks ([23]), providing more realistic and relevant measures of executive functions. Furthermore, VR offers customizability, allowing assessments to be tailored to the specific needs of diverse populations ([15]). This ensures that evaluations reflect real-world challenges and cognitive demands. These findings demonstrate the potential of VR technology in clinical and research contexts, offering a practical and accurate alternative to traditional methods. Moreover, VR can capture additional metrics beyond traditional measures of cognitive function, such as task scores or reaction times. For example, it can track arm movements and speed when using controllers, which have been reported to be closely related to cognitive function from the perspective of embodied cognition theory. These motor-related metrics can play a supportive role in assessing cognitive functions ([27]), suggesting that future VR systems should incorporate a variety of indicators to provide a more comprehensive assessment of cognitive abilities.

Despite the strengths of this study, several limitations must be acknowledged. First, the limited number of studies prevented sensitivity analyses for all subcomponents of executive function, which reduced the robustness of the findings. Second, some studies had small sample sizes, limiting the generalizability of the results. Third, although publication bias was reduced through sensitivity analysis, further studies with more rigorous methodologies are needed to validate these findings. Lastly, the study observed high heterogeneity, indicating variability in the results. Future research should aim to investigate additional factors contributing to this variability and focus on enhancing the reliability of VR-based assessments across diverse populations and contexts.

In conclusion, this meta-analysis examined the concurrent validity of VR-based executive function assessments compared to traditional paper-and-pencil tests by analyzing the subcomponents of executive function, including cognitive flexibility, attention, and inhibition. The findings revealed significant correlations between VR-based assessments and traditional measures across all subcomponents, supporting the validity of VR tools for evaluating executive function. However, the high heterogeneity observed across studies underscores the need for further research to standardize VR-based assessments and ensure their reliability across different settings.

## Figures and Tables

**Figure 1 jintelligence-12-00108-f001:**
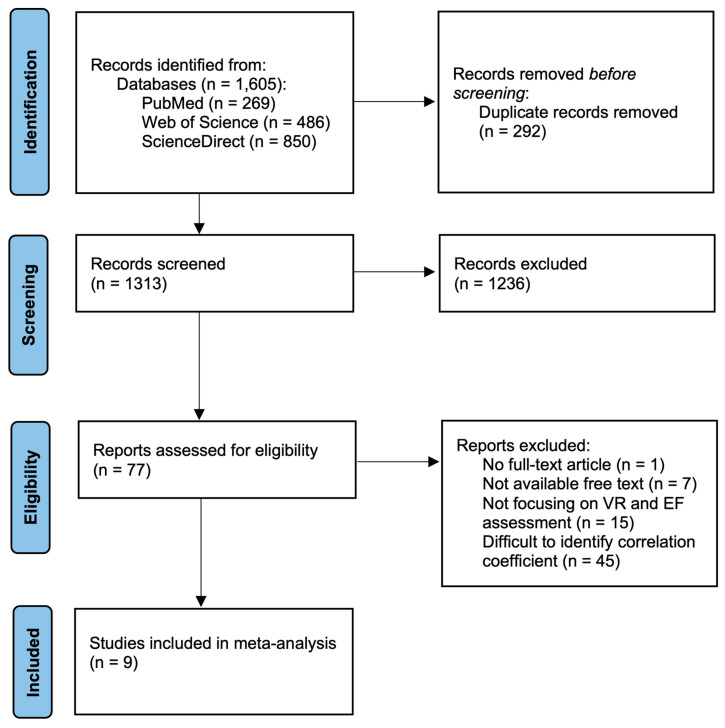
Flow chart of the study selection process.

**Figure 2 jintelligence-12-00108-f002:**
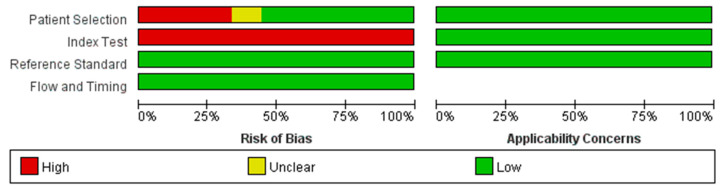
Quality Assessment of Diagnostic Accuracy Studies 2 (QUADAS-2) assessment of included studies.

**Figure 3 jintelligence-12-00108-f003:**
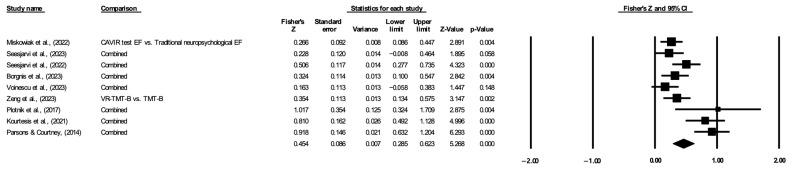
Forest plot of overall executive function. The lines with solid square represent the effect sizes for each study. The diamond symbol represents the pooled effect size and 95% confidence interval ([19]; [29]; [28]; [2]; [32]; [35]; [25]; [16]; [24]).

**Figure 4 jintelligence-12-00108-f004:**
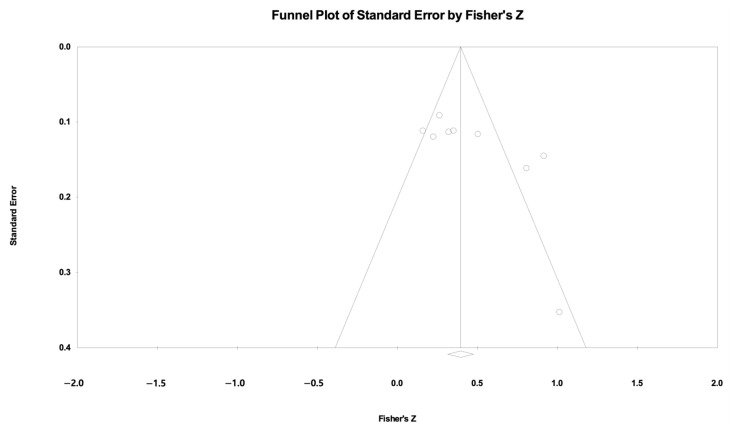
Funnel plot of overall executive function. The dots represent individual studies in a funnel plot. The diamond represents the observed effect size.

**Figure 5 jintelligence-12-00108-f005:**
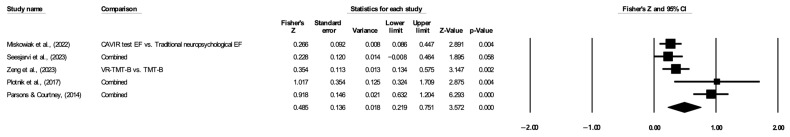
Forest plot of executive function of sensitivity analysis. The lines with solid square represent the effect sizes for each study. The diamond symbol represents the pooled effect size and 95% confidence interval ([19]; [29]; [35]; [25]; [24]).

**Figure 6 jintelligence-12-00108-f006:**
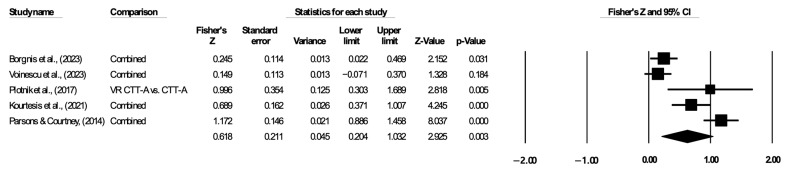
Forest plot of executive function on attention. The lines with solid square represent the effect sizes for each study. The diamond symbol represents the pooled effect size and 95% confidence interval ([2]; [32]; [25]; [16]; [24]).

**Figure 7 jintelligence-12-00108-f007:**
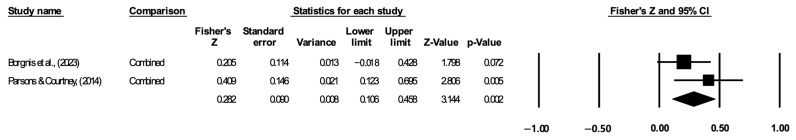
Forest plot of executive function on inhibition. The lines with solid square represent the effect sizes for each study. The diamond symbol represents the pooled effect size and 95% confidence interval ([2]; [24]).

**Figure 8 jintelligence-12-00108-f008:**
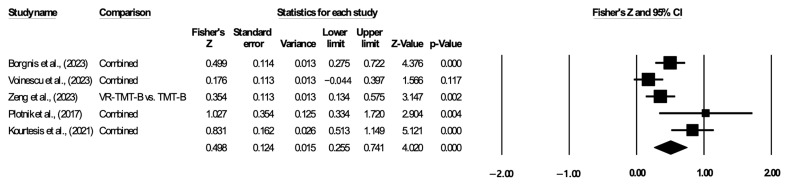
Forest plot of executive function on cognitive flexibility. The lines with solid square represent the effect sizes for each study. The diamond symbol represents the pooled effect size and 95% confidence interval ([2]; [32]; [35]; [25]; [16]).

**Table 1 jintelligence-12-00108-t001:** The main characteristics of the included studies.

Author and Year	Mean Age (Years)	% of Male Participants	N	Clinical Status of the Sample	Characteristics of VR-Based Assessments	Outcome Measure
([19])	29.92	44.63	121	Healthy and clinical (MD, PSD)	The CAVIR (Cognition Assessment in Virtual Reality): an immersive VR test of daily life cognitive functions in an interactive VR kitchen scenario.	TMT-B, CANTAB, Fluency test
([29])	11	59.72	72	Healthy children	The EPELI (Executive Performance of Everyday Living) requires participants to execute multiple tasks from memory by navigating a virtual home environment and interacting with pertinent target objects, while simultaneously keeping track of time and disregarding irrelevant distracting objects and events.	BRIEF GEC
([28])	ADHD: 10 year 4 month, Healthy: 10 year 9 month	82.89	76	Healthy and clinical (ADHD)	The EPELI (Executive Performance of Everyday Living) requires participants to execute multiple tasks from memory by navigating a virtual home environment and interacting with pertinent target objects, while simultaneously keeping track of time and disregarding irrelevant distracting objects and events.	BRIEF
([2])	66.94	41.25	80	Healthy and clinical (PD)	The EXIT 360° (EXecutive-functions Innovative Tool 360°): an innovative 360° tool designed for an ecologically valid and multi-component evaluation of executive functioning. Participants are immersed in 360° household environments, where they must complete seven everyday subtasks to simultaneously and rapidly assess various aspects of executive functioning.	PMR, AM, FAB, VF, DS, TMT-A, TMT-B, TMT-BA, ST-E, ST-T
([32])	32.4	40.2	82	Healthy and clinical (depression, anxiety)	The Nesplora Aquarium evaluates attention and executive functions in adults, based on the Continuous Performance Test (CPT) paradigm. In this VR environment, participants are instructed to focus on the main tank in the aquarium while CPTs are integrated into the experience.	TMT-A, TMT-B
([35])	47.74	49.7	82	Clinical (cancer)	The VR cognitive assessment: a virtual outdoor scenario was developed to assess executive function. This scenario includes the Trail Making Test-B (TMT-B) for assessing executive functioning.	TMT-B
([25])	37.1	81.82	11	Healthy	The VR CTT (Virtual Reality Color Trails Test): the participant’s performance, traditionally recorded with a pen-and-pencil, is now captured using a marker attached to the tip of a short pointing stick held by the participant.	TMT-A, TMT-B, TMT-BA
([16])	29.15	48.78	41	Healthy	The VR-EAL (Virtual Reality Everyday Assessment Lab) evaluates executive functioning, including planning and multitasking, within a realistic immersive VR scenario lasting around 60 min. Planning ability is assessed by having participants draw their route around the city on a 3D interactive board. Multitasking is examined through a cooking task, where participants prepare and serve breakfast and place a chocolate pie in the oven.	BADS-Key Search, CTT-1, 2
([24])	25.58	75	50	Healthy	The VR-PASAT (Virtual Reality-Paced Auditory Serial Addition Test) concentrates on the detailed analysis of neurocognitive testing within a virtual gaming environment to evaluate attention processing and executive functioning while navigating a virtual city.	PASAT-200, D-KEFS Color-Word Interference Test (Inhibition, Inhibition/Switching)

MD: mood disorders; PSD: psychosis spectrum disorders; TMT: Trail Making Test; CANTAB: Cambridge Neuropsychological Test Automated Battery; BRIEF GEC: Behavior Rating Inventory of Executive Function Global Executive Composite; ADHD: attention-deficit/hyperactivity disorder; PD: Parkinson’s disease; PMR: progressive matrices of raven; AM: attentive matrices; FAB: frontal assessment battery; VF: verbal fluency; DS: digit span; ST-E: Stroop test-errors; ST-T: Stroop test-time; BADS: Behavioral Assessment of the Dysexecutive Syndrome; CTT: Color Trails Test; D-KEFS: Delis–Kaplan Executive Function System.

**Table 2 jintelligence-12-00108-t002:** Outcome measures by executive function sub-elements.

Executive Function Sub-Elements	Outcome Measures
Attention	TMT-A, CTT-1, AM, PASAT-200
Inhibition	ST-E, ST-T, D-KEFS Color-Word Interference Test (Inhibition, Inhibition/Switching)
Cognitive flexibility	TMT-B, TMT-BA, CTT-2

TMT: Trail Making Test; CTT: Color Trails Test; AM: attentive matrices; ST-E: Stroop test-errors; ST-T: Stroop test-time; D-KEFS: Delis–Kaplan Executive Function System.

## Data Availability

All data are publicly available.

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
