# Peer review of "Concurrent Validity of Virtual Reality-Based Assessment of Executive Function: A Systematic Review and Meta-Analysis"

_jintelligence, 2024, doi:10.3390/jintelligence12110108_

Round 1

Reviewer 1 Report

Comments and Suggestions for Authors

This meta-analysis wants to investigate the concurrent validity between VR-based assessment of executive function and traditional neuropsychological assessment. The study is quite interesting. However, it requires significant revision before it can be published.

Introduction

In the whole introduction, please pay attention to the references. In some cases, there are more recent works available in the literature. Specifically, the concept of executive functions has always attracted the attention of many researchers and is continuously evolving. For example, the definition of executive functions has been extensively revised compared to the references cited by the authors in the text.

Moreover, please, ensure that each concept is supported by at least one reference.

In some cases, the authors arrive at 'conclusions' using terms like 'thus' and 'therefore' that are not always easily understandable considering what has been stated previously (and without reference). For example, “Thus, evaluating and monitoring executive functions is crucial as it enables the prediction of their ability to perform their occupation effectively across different age groups”. Please check that the text is smooth and easily comprehensible.

For a broader audience, it would be important to detail the difference between convergent validity and concurrent validity. Additionally, what does this study add compared to the first one??

Methods

Literature search: 4 databases?

Has this review been registered in any registry such as PROSPERO? Have the authors checked that there are no similar reviews? For example, there is a systematic review with the same search terms in the literature (10.3389/fpsyg.2022.833136). Are the papers found the same? Please, detail.

Results

The authors combine different characteristics in terms of age, target population, and the tool used. Don’t they think these differences could impact the synthesis of the results? Furthermore, the VR tools used are different. Are they sure that comparing paper-and-pencil tests in VR format with completely new tests can yield the same results?

Study quality: Please provide more details. The figure shows many values that have not been described

Results of the Meta-analysis: “In this study, executive function was subdivided into the following subcomponents: attention, inhibition, and cognitive flexibility.” Why did the authors choose these three subcomponents of executive functioning?

I would like to know if the authors of the studies considered in the paper had already conducted a concurrent validity analysis of their tests. If so, are the results the same?

Discussion:

The discussion presents some critical points:

-        The authors should not only report the results that have already emerged in the previous section of the discussion. Do not repeat information that has already been stated

-        Furthermore, the authors state “This study confirmed that VR-based assessment of executive function demonstrates considerable validity compared to traditional paper-and-pencil assessments”' when they find few significant correlations with the neuropsychological tests.

-        In some cases, the authors cite studies that do not pertain to executive functions. Since, as mentioned earlier, there are many studies on executive functions, the authors should discuss their results using appropriate studies.

Abstract

The conclusion of the abstract does not align with the discussion of the paper.

Comments on the Quality of English Language

good

Author Response

Introduction

#1. In the whole introduction, please pay attention to the references. In some cases, there are more recent works available in the literature. Specifically, the concept of executive functions has always attracted the attention of many researchers and is continuously evolving. For example, the definition of executive functions has been extensively revised compared to the references cited by the authors in the text.

Moreover, please, ensure that each concept is supported by at least one reference.

Response #1: Thank you for your valuable feedback. While we acknowledge that the division of executive functions into three primary subcomponents—such as cognitive flexibility, attention, and inhibition—follows a well-established model, we did not include more recent research on this specific categorization as it remains consistent with earlier models. However, in the revised manuscript, we have emphasized, based on the latest studies, that these subcomponents are separable yet interrelated. This emphasis aligns with the evolving understanding of executive function, highlighting the importance of examining these distinct yet connected components. Additionally, we have ensured that each concept is supported by at least one relevant and current reference.

We have added this information in the Introduction section (“Many studies have identified three key subcomponents of executive function: working memory, inhibition, and cognitive flexibility (Diamond 2013; Miyake et al. 2000). However, Miyake et al.’s (2000) model was not intended to be comprehensive, acknowledging the possibility of additional separable subcomponents. Recent studies have reinforced the importance of viewing executive function as distinct subcomponents rather than a single unified function. Neurological evidence from large-scale brain networks supports this perspective, showing that while these components are anatomically and functionally distinct, they interact within integrated circuits to support complex cognitive tasks (Friedman and Robbins 2022; Zelazo and Carlson 2023). This approach provides a more refined framework for both theoretical models and practical assessments of executive function.”)

#2. In some cases, the authors arrive at 'conclusions' using terms like 'thus' and 'therefore' that are not always easily understandable considering what has been stated previously (and without reference). For example, “Thus, evaluating and monitoring executive functions is crucial as it enables the prediction of their ability to perform their occupation effectively across different age groups”. Please check that the text is smooth and easily comprehensible.

Response #2: Thank you for your valuable feedback regarding the clarity of the text. We have revised the sentence to improve the flow and ensure better comprehension. Specifically, we restructured the statement to clarify that impairments in executive function can negatively impact academic performance, daily living activities, and the ability to manage illness. Therefore, monitoring and evaluating executive functions is crucial. This revised flow ensures the logical connection between the problem and the need for monitoring executive functions.

We have added this information in the Introduction section (“Executive functions, considered higher-order cognitive abilities, are essential for managing tasks across various age groups and clinical populations. Impairments in executive functions weaken academic performance and reduce the ability to carry out independent activities in daily life (Cortés Pascual et al. 2019; Lau et al. 2015). Impairments in executive functions also negatively affect disease management, as individuals may fail to recognize the severity of their health problems (Ibrahim et al. 2017). Therefore, assessing and monitoring executive functions is vital in predicting whether individuals can effectively manage everyday tasks and activities related to education and daily life.”)

#3. For a broader audience, it would be important to detail the difference between convergent validity and concurrent validity. Additionally, what does this study add compared to the first one?

Response #3: Thank you for your feedback. The primary difference between our study and prior research is our approach to examining executive function. Our study breaks down executive function into subcomponents, providing a more detailed analysis compared to previous studies. This distinction in perspective is where the key difference lies, and we have highlighted this in the revised manuscript. Therefore, we did not elaborate on the distinction between convergent and concurrent validity.

Additionally, for a broader audience, we have also included an explanation of concurrent validity to clarify its importance in our research. We have added this information in the Introduction section (“Concurrent validity measures the extent to which a new test correlates with an established one when both are administered simultaneously (Godwin et al. 2013). Our analysis focuses on whether VR-based assessments yield comparable results to traditional methods in evaluating executive function.”)

Methods

#4. Literature search: 4 databases?

Response #4: We apologize for the confusion in the initial submission. The correct number of databases used in the literature search is three, not four. These databases include PubMed, Web of Science, and ScienceDirect. We have corrected this information in the manuscript to accurately reflect the search methodology. At the time of our search, we limited the databases to these three due to their accessibility and relevance to research on VR-based assessments and executive function. Additionally, previous studies (NEGUÅ¢ et al. 2015) have also utilized three databases for their literature searches, supporting our approach to the selection of sources for this review.

#5. Has this review been registered in any registry such as PROSPERO? Have the authors checked that there are no similar reviews? For example, there is a systematic review with the same search terms in the literature (10.3389/fpsyg.2022.833136). Are the papers found the same? Please, detail.

Response #5: Thank you for your valuable comment. We confirm that this review has been registered in the PROSPERO registry under the ID CRD42024507500. We have added this information in the Methods section (“This systematic review and meta-analysis was conducted in accordance with the Preferred Reporting Items for Systematic Review and Meta-Analyses (PRISMA) guidelines. This study was registered with PROSPERO (ID: CRD42024507500).”)

Regarding the mention (10.3389/fpsyg.2022.833136), we have thoroughly checked it, and while there is some overlap in the focus on executive function assessment, there are two key differences between the mentioned one and ours. The cited article takes a broader approach to executive function assessments in general, whereas our study specifically focuses on the concurrent validity of VR-based executive function assessments compared to traditional neuropsychological tools, providing a more detailed analysis of the technical methods used in the assessments. Additionally, our study conducts an independent analysis of the subcomponents of executive function (e.g., cognitive flexibility, attention, inhibition) rather than treating executive function as a single construct, while the mentioned study adopts a more generalized approach without offering in-depth analysis of these subcomponents. Several studies have highlighted that executive functions consist of separable but interrelated components, which supports our decision to conduct a more nuanced analysis of these subcomponents (Doebel 2020; Friedman and Robbins 2022).

We have added this information in the Discussion section (“Our findings are consistent with previous studies that demonstrated correlations between VR-based assessments and traditional neuropsychological evaluations (NEGUÅ¢ et al. 2015). Additionally, the results of the nine studies considered in this paper align with our findings, as most reported significant correlations. However, while prior studies treated executive function as a single construct without providing an in-depth analysis of its subcomponents (Borgnis et al. 2022; NEGUÅ¢ et al. 2015), our research offers independent analyses of key subcomponents, such as cognitive flexibility, attention, and inhibition. This distinction is important, as multiple studies have indicated that executive function consists of separable but interrelated components (Doebel 2020; Friedman and Robbins 2022). Therefore, our study’s focus on analyzing these subcomponents independently adds meaningful insights.”)

Results
#6. The authors combine different characteristics in terms of age, target population, and the tool used. Don’t they think these differences could impact the synthesis of the results? Furthermore, the VR tools used are different. Are they sure that comparing paper-and-pencil tests in VR format with completely new tests can yield the same results?

Response #6: Thank you for raising these important concerns. Before addressing this comment, we would like to inform you of an important update. During the process of revising the manuscript, we realized that a significant error had been made in the analysis, specifically with the effect direction in the CMA program being set to "Auto." This oversight impacted the initial results. After correcting this mistake, we were able to reassess the data and found that there is indeed concurrent validity between traditional assessments and VR-based assessments across overall executive function and its subcomponents. This correction has added further meaning and validity to our findings, confirming the robustness of VR-based assessments in measuring executive function.

We understand that variations in age, clinical status, and the properties of VR tools could potentially influence the synthesis of the results. To address this, we conducted detailed subgroup analyses to explore whether these factors moderated the relationship between VR-based assessments and traditional neuropsychological assessments. We performed a subgroup analysis to determine whether the age of the sample (children vs. adults) moderated the correlation between VR-based and traditional assessments. The analysis for overall executive function indicated that age did not significantly moderate the results. This suggests that the age differences did not substantially affect the validity of the comparisons between VR-based and traditional assessments.

We have added this information in the Results section (“We conducted a subgroup analysis to determine whether the age of the sample moderated the correlation between VR-based assessments and traditional assessments. This analysis was performed for overall executive function only. The results indicated that age did not moderate the relationship between VR-based and traditional assessments in the overall executive function analysis (Q-value(df) = 0.305(1), p = .581).”).

Another subgroup analysis was conducted to examine the moderating effect of clinical status. For both overall executive function and cognitive flexibility, clinical status did not moderate the results. However, in the case of attention, clinical status was found to be a significant moderator, indicating that clinical populations showed different patterns of correlation compared to non-clinical populations.

We have added this information in the Results section (“We conducted a subgroup analysis to determine whether the clinical status of the sample moderated the correlation between VR-based assessments and traditional assessments. The analysis was performed for overall executive function, as well as the subcomponents of attention and cognitive flexibility. The results indicated that clinical status did not moderate the relationship between VR-based and traditional assessments in the overall executive function analysis (Q-value(df) = 4.18(2), p = .124) or the cognitive flexibility analysis (Q-value(df) = 5.671(2), p = .059). However, a moderating effect of clinical status was found in the attention analysis (Q-value(df) = 17.215(1), p < .001).”)

We also examined whether the properties of VR-based assessments (adaptations of traditional assessments vs. newly developed VR tasks) moderated the results. Subgroup analysis showed that VR properties did not significantly moderate the relationship between VR-based and traditional assessments for overall executive function, attention, or cognitive flexibility.

We have added this information in the Results section (“We conducted a subgroup analysis to determine whether the properties of VR-based assessments moderated the correlation between VR-based assessments and traditional assessments. The analysis was performed on overall executive function, as well as the subcomponents of attention and cognitive flexibility. A total of nine studies were included in the executive function analysis, while five studies each were included for the attention and cognitive flexibility analyses. The results indicated that VR-based assessment properties did not moderate the relationship between VR-based and traditional assessments in the overall executive function analysis (Q-value(df) = 0.364(1), p = .546). Similarly, no moderating effect of VR properties was found in the attention analysis (Q-value(df) = 0.307(1), p = .579) or the cognitive flexibility analysis (Q-value(df) = 1.508(1), p = .219).”)

#7. Study quality: Please provide more details. The figure shows many values that have not been described

Response #7: Thank you for your valuable feedback. We have expanded the discussion of study quality in the revised manuscript, particularly by adding further details regarding the index test, flow and timing, and reference standard domains.

We have added this information in the Results section (“The risk of bias in the index test was rated high across all studies because none of the VR-based tools included predefined thresholds for evaluation. While the absence of these predefined thresholds contributed to the high-risk rating in the index test, it was considered not critical for this study as it did not affect the main results of the meta-analysis. For flow and timing and reference standard, most studies adhered to acceptable practices.”)

Despite these limitations, all studies were initially included for the analysis of overall executive function. A sensitivity analysis was then conducted by excluding low-quality studies to assess the robustness of the overall results. The results remained statistically significant, suggesting that the findings were not unduly influenced by the low-quality studies. For the analysis of executive function subcomponents, all available studies, including the low-quality ones, were retained to ensure sufficient statistical power. This allowed us to present a comprehensive evaluation of the concurrent validity of VR-based assessments.

We have added this information in the Results section (“Despite these limitations, all studies were initially included for analysis of overall executive function. A sensitivity analysis was then conducted by excluding low-quality studies to assess the robustness of the overall results. For the analysis of executive function subcomponents, all available studies, including the low-quality ones, were used to ensure sufficient statistical power.”)

#8. Results of the Meta-analysis: “In this study, executive function was subdivided into the following subcomponents: attention, inhibition, and cognitive flexibility.” Why did the authors choose these three subcomponents of executive functioning?

Response #8: Thank you for your valuable question. The selection of attention, inhibition, and cognitive flexibility as subcomponents of executive function in this meta-analysis was not arbitrary. Instead, these three subcomponents were chosen because they were the most commonly assessed factors across the selected studies. Specifically, at least two of the included studies evaluated these subcomponents using similar or comparable paper-and-pencil assessment tools. This allowed us to group the studies and make meaningful comparisons between VR-based assessments and traditional assessments. By focusing on these three subcomponents, we ensured that the analysis maintained a consistent basis across the selected studies, thereby enhancing the reliability and validity of the comparisons made in the meta-analysis.

We have added this information in the Discussion section (“The subcomponents of inhibition, attention, and cognitive flexibility were analyzed separately to gain a more detailed understanding of how VR-based assessments relate to traditional neuropsychological tools. These subcomponents were chosen because at least two studies assessed each using similar paper-and-pencil tests, allowing for meaningful comparisons across studies.”)

#9. I would like to know if the authors of the studies considered in the paper had already conducted a concurrent validity analysis of their tests. If so, are the results the same?

Response #9: Thank you for your question regarding the concurrent validity analysis conducted in the included studies. All nine studies in our meta-analysis performed a concurrent validity analysis of their tests. Among these, four studies (Miskowiak et al. 2022; Seesjärvi et al. 2022; Zeng et al. 2023; Kourtesis et al. 2021) reported significant correlations in all their results, three studies (Borgnis et al. 2023; Plotnik et al. 2017; Parsons and Courtney 2014) found significant correlations in most of their results, and two studies (Seesjärvi et al. 2023; Voinescu et al. 2023) observed significant correlations in about half or fewer of their outcomes. These findings align with the overall results of our meta-analysis, which supports the concurrent validity of VR-based assessments of executive function when compared to traditional methods.

We have added this information in the Discussion section (“Additionally, the results of the nine studies considered in this paper align with our findings, as most reported significant correlations.”)

Discussion

#10. The authors should not only report the results that have already emerged in the previous section of the discussion. Do not repeat information that has already been stated

Response #10: We appreciate the reviewer’s suggestion to avoid repeating results in the discussion section. In the revised manuscript, we have focused the discussion on interpreting the results, exploring their significance, and comparing them with findings from existing literature. Instead of restating the results, we have emphasized the implications of the findings, addressed the study's limitations, and highlighted directions for future research.

#11. Furthermore, the authors state “This study confirmed that VR-based assessment of executive function demonstrates considerable validity compared to traditional paper-and-pencil assessments”' when they find few significant correlations with the neuropsychological tests.

Response #11: Thank you for your feedback. The study’s findings have been revised, and we have considered your comments during the update. The revised results now demonstrate statistically significant correlations across all subcomponents of executive function, including cognitive flexibility, attention, and inhibition. Consequently, the statement regarding the considerable validity of VR-based assessments compared to traditional paper-and-pencil assessments is supported by the updated data. We have ensured that the revised manuscript reflects these findings accurately without overstating the results.

#12. In some cases, the authors cite studies that do not pertain to executive functions. Since, as mentioned earlier, there are many studies on executive functions, the authors should discuss their results using appropriate studies.

Response #12: Thank you for bringing this to our attention. In the revised manuscript, we have carefully ensured that all references are specifically related to executive functions, and any irrelevant citations have been replaced with studies that align more closely with the focus of the paper. Additionally, we will discuss the results in the context of established findings in the executive function literature, ensuring that the discussion remains relevant and supported by up-to-date research. However, I have used EndNote to revise the references, and due to technical issues, tracking was not applied to the reference list.

Abstract
#13. The conclusion of the abstract does not align with the discussion of the paper.

Response #13: Thank you for your feedback. The results of the study have been updated, and we have ensured that the conclusion of the abstract now aligns with the discussion section. The revised abstract accurately reflects the findings of the study, including the significant correlations between VR-based and traditional assessments across all subcomponents of executive function. We have also made sure that the implications and conclusions drawn in the discussion are consistent with those presented in the abstract.

We have added this information in the Abstract section (“The results revealed statistically significant correlations between VR-based assessments and traditional measures across all subcomponents. Sensitivity analyses confirmed the robustness of the findings, even when low-quality studies were excluded. These results support the use of VR-based assessments as a valid alternative to traditional methods for evaluating executive function. The study highlights the potential of VR-based assessments as a valid alternative to traditional methods, emphasizing the need to address variability in executive function subcomponents and integrate diverse cognitive and motor metrics for greater ecological validity.”)

Reviewer 2 Report

Comments and Suggestions for Authors

The author wrote the manuscript entitled, “Concurrent validity of virtual reality-based assessment of executive function: A systematic review and meta-analysis.” It is very interesting to read about the concurrent validity of VR assessment of executive function.

However, I have some comments for the improvement of the manuscript.

Abstract

The abstract is not clear enough on how concurrent validity was assessed in the study. Moreover, the brief findings should be framed more clearly in terms of concurrent validity for all subcomponents of executive function, including attention and inhibition.

Introduction

The authors focused on the concurrent validity of VR assessment. However, the definition of concurrent validity and its importance were still missing.

In describing the flow diagram of PRISMA, it would be better the total number of documents downloaded.

Methods

Literature Search

The authors limited the search databases to three, such as PubMed, WoS, and ScienceDirect. And they included only nine studies to be reviewed. This may cause missing relevant studies. This limitation could resist the comprehensiveness of the review and meta-analysis, reducing the study’s generalizability.

Therefore, I highly recommend expanding the search for additional sources such as PsycINFO, Scopus, Cochrane, etc.

Moreover, the authors can use more additional keywords (immersive technology or augmented reality instead of VR keyword) to capture various aspects of executive function (e.g., cognitive flexibility, working memory, inhibition, VR assessment).

Timeframe

The authors also limited the timeframe of downloaded documents from 2013-2023. But they did not justify it. Now, the included documents are few (just 9 papers). Why can’t the authors expand the timeframe for more documents or information?

Eligibility & Screening

The inclusion criteria focus on VR assessments and executive function but exclude case studies and reviews, which might provide valuable insights into good VR applications and theoretical discussions. Could you please justify it as well?

Screening and Data Extraction Process

The authors described that the two reviewers screened the titles and abstracts, with discrepancies resolved by a third reviewer. However, they do not provide information on inter-rater reliability or the specific method used to resolve discrepancies.

Findings

Quality assessment

The use of the QUADAS-2 tool for quality assessment is appropriate, but there is no detailed explanation of how the authors handled studies that did not meet all quality criteria.

Therefore, I recommend including more details about how they assessed the quality of studies and how lower-quality studies were weighted or treated in the analysis. E.g., were any studies excluded due to poor quality, and if so, why? Were sensitivity analyses performed to account for the potential impact of study quality on the results?

High Heterogeneity

The authors mentioned that the heterogeneity of the studies was high (I2 > 50%), but they did not explain the sources of this heterogeneity or how it was addressed beyond applying a random-effects model. Therefore, it would be better to conduct subgroup analyses or meta-regression to explore the sources of heterogeneity (e.g., by population, VR assessment type, clinical vs non-clinical samples).

Publication Bias Assessment

The authors used Funnel plots and Egger’s test to assess publication bias, the authors do not provide details about the results of these tests or how publication bias was handled in the meta-analysis.

Moreover, the funnel plot and Egger’s test for attention and cognitive flexibility indicate no publication bias, inhibition was not analyzed due to an insufficient number of studies. This incomplete assessment limits the overall conclusions about bias across the study domains.

Discussion

The authors wrote the discussion part by describing the summary of the results. I mean the authors did not discuss the findings relating to the reviewed literature and aims. If necessary, the authors need to give reasons or explanations about their findings. For example,

·       The authors repeatedly mentioned a lack of significant correlations between VR-based assessments and traditional tests, particularly for attention and inhibition, but they did not provide a thorough explanation of these findings and the reasons behind them.

·       The risk of bias, particularly in patient selection, is described but not discussed in relation to how it might impact findings on attention, inhibition, and cognitive flexibility.

·       The authors described high heterogeneity but did not adequately discuss what factors contributed to this heterogeneity.

·       Although the discussion briefly discussed the potential benefits of VR, it still lacks of discussion on the specific advantages relating to real-world applications.

·       The authors discussed VR adaptations of traditional tests but failed to critically assess whether these adaptations might have influenced the results.

·       Although the authors mentioned the need for further refinement of VR assessments, they did not provide clear recommendations for future research.

·       Moreover, they did not discuss well about the implications of the study.

To conclude, I hope these comments are useful for improving the quality of the manuscript. Thanks.

Comments on the Quality of English Language

Minor editing of English language required.

Author Response

Abstract
#1. The abstract is not clear enough on how concurrent validity was assessed in the study. Moreover, the brief findings should be framed more clearly in terms of concurrent validity for all subcomponents of executive function, including attention and inhibition.

Response #1: Thank you for your valuable feedback. Before addressing this comment, we would like to inform you of an important update. During the process of revising the manuscript, we realized that a significant error had been made in the analysis, specifically with the effect direction in the CMA program being set to "Auto." This oversight impacted the initial results. After correcting this mistake, we were able to reassess the data and found that there is indeed concurrent validity between traditional assessments and VR-based assessments across overall executive function and its subcomponents. This correction has added further meaning and validity to our findings, confirming the robustness of VR-based assessments in measuring executive function.

We have revised the abstract to provide a clearer explanation of how concurrent validity was assessed in the study. Specifically, we added a statement to clarify that the effect size for overall executive function was evaluated, and the subcomponents (cognitive flexibility, attention, and inhibition) were categorized based on the assessment tools used. The effect size for each subcomponent was analyzed separately to offer a more detailed understanding of their relationships with traditional measures. We have added this information in the Abstract section (“The effect size for overall executive function was assessed, with subcomponents categorized based on the specific assessment tools used in the studies. The effect size for each subcomponent—cognitive flexibility, attention, and inhibition—was then analyzed to provide a more detailed understanding of their relationships with traditional measures.”)

Additionally, we updated the results section in the abstract to reflect the findings for all subcomponents of executive function, including attention and inhibition, ensuring that the concurrent validity is framed more clearly for each aspect. We have added this information in the Abstract section (“The results revealed statistically significant correlations between VR-based assessments and traditional measures across all subcomponents. Sensitivity analyses confirmed the robustness of the findings, even when low-quality studies were excluded.”)

Introduction

#2. The authors focused on the concurrent validity of VR assessment. However, the definition of concurrent validity and its importance were still missing.

Response #2: We have added a clear definition of concurrent validity in the revised manuscript to address this concern. We have added this information in the Introduction section (“Concurrent validity measures the extent to which a new test correlates with an established one when both are administered simultaneously (Godwin et al. 2013). Our analysis focuses on whether VR-based assessments yield comparable results to traditional methods in evaluating executive function.”)

#3. In describing the flow diagram of PRISMA, it would be better the total number of documents downloaded.

Response #3: We have reviewed the manuscript and ensured that the total number of downloaded documents is clearly stated in the main text. Additionally, we have updated Figure 1 to include the total number of documents downloaded, providing a clearer overview of the screening process following PRISMA guidelines.

Methods

Literature Search
#4. The authors limited the search databases to three, such as PubMed, WoS, and ScienceDirect. And they included only nine studies to be reviewed. This may cause missing relevant studies. This limitation could resist the comprehensiveness of the review and meta-analysis, reducing the study’s generalizability.

Therefore, I highly recommend expanding the search for additional sources such as PsycINFO, Scopus, Cochrane, etc.

Response #4: Thank you for your suggestion to expand the search to additional databases such as PsycINFO, Scopus, and Cochrane. However, at the time of our search, we limited the databases to PubMed, Web of Science (WoS), and ScienceDirect due to their accessibility and relevance to research on VR-based assessments and executive function. These databases were chosen for their broad coverage of scientific and clinical research, including neuropsychological assessments and VR tools. Furthermore, previous studies (NEGUÅ¢ et al. 2015) have also utilized three databases for their literature searches, supporting the appropriateness of our approach.

While expanding the search could potentially yield more studies, doing so now would alter the search timeline and affect the consistency of the studies initially analyzed. Furthermore, it may necessitate a re-evaluation of the results, which is beyond the scope of this current study. Nonetheless, we acknowledge this as a limitation and recognize that expanding the search to include additional databases like PsycINFO and Scopus could enhance the comprehensiveness of future reviews.

#5. Moreover, the authors can use more additional keywords (immersive technology or augmented reality instead of VR keyword) to capture various aspects of executive function (e.g., cognitive flexibility, working memory, inhibition, VR assessment).

Response #5: Thank you for your suggestion to include additional keywords such as "immersive technology" or "augmented reality" to capture a broader range of research on executive function. After reviewing relevant VR literature, we found that using the keyword "virtual reality" alone provided a comprehensive and extensive search, encompassing studies on executive function, including subcomponents like cognitive flexibility, working memory, and inhibition.

Given that "virtual reality" is the primary term used in most research focusing on immersive environments for neuropsychological and cognitive assessments, we believe that this keyword sufficiently covers the literature relevant to our study’s goals. Many studies using VR tools, even if they incorporate elements of immersive technology or augmented reality, tend to fall under the broader category of virtual reality. As a result, we believe that using "virtual reality" in our search strategy ensures a broad review.

While we acknowledge the value of exploring other related technologies, such as augmented reality, our review focused specifically on tools that explicitly use virtual reality for executive function assessment. Incorporating keywords like "immersive technology" or "augmented reality" could help expand the scope in future studies, but we believe that "virtual reality" alone was sufficient for capturing the current literature on VR-based cognitive assessments in this study.

Timeframe

#6. The authors also limited the timeframe of downloaded documents from 2013-2023. But they did not justify it. Now, the included documents are few (just 9 papers). Why can’t the authors expand the timeframe for more documents or information?

Response #6: Thank you for your suggestion to expand the timeframe of the document search. The decision to limit the search to studies published between 2013 and 2023 was based on two key factors.

First, it is common practice in systematic reviews to focus on 10 years to ensure that the literature reflects the most recent advancements in the field. This approach allows us to review the most relevant and up-to-date studies without including older research that may no longer represent the current state of VR-based cognitive assessments.

Additionally, a previous systematic review has already covered studies up to September 2014, focusing on earlier research in this field. Expanding the timeframe beyond 2013 would likely result in significant overlap with that earlier review.

We aimed to build upon those findings by focusing on recent developments since 2013, which capture the evolution of VR technology in executive function assessments over the past decade. Considering these factors, we believe that the 2013-2023 period is appropriate for this review and enables us to gather the most relevant and up-to-date literature in this field.

Eligibility & Screening

#7. The inclusion criteria focus on VR assessments and executive function but exclude case studies and reviews, which might provide valuable insights into good VR applications and theoretical discussions. Could you please justify it as well?

Response #7: We appreciate the reviewer's concern regarding the exclusion of case studies and review articles. We excluded case studies and reviews because, in a meta-analysis, it is standard practice to focus on original articles that provide empirical data for statistical analysis. Reviews are typically excluded because they do not contribute original data but instead summarize existing studies. Similarly, case studies were excluded due to the lack of sufficient comparative data, making them unsuitable for inclusion in a meta-analysis. This approach ensures the robustness of our findings by relying on studies that meet the necessary criteria for statistical comparison and synthesis.

Additionally, studies that did not report correlation coefficients or where such values were difficult to identify were excluded. The main objective of this study was to evaluate concurrent validity through measurable correlations between VR-based and traditional assessments. By focusing on studies that reported this data, we aimed to ensure the robustness and reliability of the meta-analysis.

We have added this information in the Methods section (“Additionally, studies were excluded if they made it difficult to identify the correlation coefficients between VR-based assessments and traditional paper-and-pencil tests.”)

Screening and Data Extraction Process

#8. The authors described that the two reviewers screened the titles and abstracts, with discrepancies resolved by a third reviewer. However, they do not provide information on inter-rater reliability or the specific method used to resolve discrepancies.

Response #8: We appreciate the reviewer's concern regarding the screening and data extraction process. In our study, two reviewers independently screened the titles and abstracts of the selected studies based on the inclusion criteria. Full-text reviews were then conducted to identify eligible studies. In cases where discrepancies occurred during the screening process, the two reviewers discussed the differences. If consensus was not reached, a third reviewer was consulted to resolve the discrepancies. This process ensured an unbiased and rigorous selection of studies.

We have added this information in the Methods section (“In cases where discrepancies arose, the two reviewers discussed the differences. If they were unable to reach a consensus, a third reviewer was consulted to resolve the disagreement.”)

Findings

Quality assessment

#9. The use of the QUADAS-2 tool for quality assessment is appropriate, but there is no detailed explanation of how the authors handled studies that did not meet all quality criteria.

Therefore, I recommend including more details about how they assessed the quality of studies and how lower-quality studies were weighted or treated in the analysis. E.g., were any studies excluded due to poor quality, and if so, why? Were sensitivity analyses performed to account for the potential impact of study quality on the results?

Response #9: We appreciate the reviewer's concern regarding how studies that did not meet all quality criteria were handled in the analysis.

In addressing lower-quality studies, sensitivity analyses provided key insights. In the initial meta-analysis, we included all 9 studies to ensure sufficient statistical power. Although the quality of these studies varied, they contributed to the overall assessment of executive function using VR-based tools. While statistical significance was found in the initial analysis, high heterogeneity and the presence of publication bias were noted.

To address the potential impact of study quality, we performed a sensitivity analysis by excluding 4 studies that did not meet certain quality criteria, such as having an unclear or high risk of bias in key areas like patient selection and index test. The sensitivity analysis, conducted on the remaining 5 high-quality studies, still demonstrated statistical significance. Although high heterogeneity remained, publication bias was no longer detected after excluding lower-quality studies.

Due to the limited number of studies available for meta-analysis, we initially retained the lower-quality studies to maintain sufficient statistical power. However, the sensitivity analysis confirmed that the results remained robust even after excluding these studies.

We have added this information in the Results section (“Despite these limitations, all studies were initially included for analysis of overall executive function. A sensitivity analysis was then conducted by excluding low-quality studies to assess the robustness of the overall results. For the analysis of executive function subcomponents, all available studies, including the low-quality ones, were used to ensure sufficient statistical power.”)

High Heterogeneity

#10. The authors mentioned that the heterogeneity of the studies was high (I2 > 50%), but they did not explain the sources of this heterogeneity or how it was addressed beyond applying a random-effects model. Therefore, it would be better to conduct subgroup analyses or meta-regression to explore the sources of heterogeneity (e.g., by population, VR assessment type, clinical vs non-clinical samples).

Response #10: Thank you for raising this important concern. We performed a subgroup analysis to determine whether the age of the sample (children vs. adults) moderated the correlation between VR-based and traditional assessments. The analysis for overall executive function indicated that age did not significantly moderate the results. This suggests that the age differences did not substantially affect the validity of the comparisons between VR-based and traditional assessments.

We have added this information in the Results section (“We conducted a subgroup analysis to determine whether the age of the sample moderated the correlation between VR-based assessments and traditional assessments. This analysis was performed for overall executive function only. The results indicated that age did not moderate the relationship between VR-based and traditional assessments in the overall executive function analysis (Q-value(df) = 0.305(1), p = .581).”)

Another subgroup analysis was conducted to examine the moderating effect of clinical status. For both overall executive function and cognitive flexibility, clinical status did not moderate the results. However, in the case of attention, clinical status was found to be a significant moderator, indicating that clinical populations showed different patterns of correlation compared to non-clinical populations.

We have added this information in the Results section (“We conducted a subgroup analysis to determine whether the clinical status of the sample moderated the correlation between VR-based assessments and traditional assessments. The analysis was performed for overall executive function, as well as the subcomponents of attention and cognitive flexibility. The results indicated that clinical status did not moderate the relationship between VR-based and traditional assessments in the overall executive function analysis (Q-value(df) = 4.18(2), p = .124) or the cognitive flexibility analysis (Q-value(df) = 5.671(2), p = .059). However, a moderating effect of clinical status was found in the attention analysis (Q-value(df) = 17.215(1), p < .001).”)

We also examined whether the properties of VR-based assessments (adaptations of traditional assessments vs. newly developed VR tasks) moderated the results. Subgroup analysis showed that VR properties did not significantly moderate the relationship between VR-based and traditional assessments for overall executive function, attention, or cognitive flexibility.

We have added this information in the Results section (“We conducted a subgroup analysis to determine whether the properties of VR-based assessments moderated the correlation between VR-based assessments and traditional assessments. The analysis was performed on overall executive function, as well as the subcomponents of attention and cognitive flexibility. A total of nine studies were included in the executive function analysis, while five studies each were included for the attention and cognitive flexibility analyses. The results indicated that VR-based assessment properties did not moderate the relationship between VR-based and traditional assessments in the overall executive function analysis (Q-value(df) = 0.364(1), p = .546). Similarly, no moderating effect of VR properties was found in the attention analysis (Q-value(df) = 0.307(1), p = .579) or the cognitive flexibility analysis (Q-value(df) = 1.508(1), p = .219).”)

Publication Bias Assessment

#11. The authors used Funnel plots and Egger’s test to assess publication bias, the authors do not provide details about the results of these tests or how publication bias was handled in the meta-analysis.

Moreover, the funnel plot and Egger’s test for attention and cognitive flexibility indicate no publication bias, inhibition was not analyzed due to an insufficient number of studies. This incomplete assessment limits the overall conclusions about bias across the study domains.

Response #11: In the initial meta-analysis of all nine studies, both the Funnel plot and Egger’s test indicated the presence of publication bias. This suggests that smaller studies with non-significant results may be underrepresented in the literature, potentially influencing the overall conclusions drawn from the included studies. After performing a sensitivity analysis that excluded four low-quality studies, the results still showed evidence of publication bias, indicating that even the higher-quality studies exhibited some degree of bias, and the potential influence of publication bias persisted despite considering study quality.

When analyzing the subcomponents of executive function separately, no evidence of publication bias was found for attention and cognitive flexibility, as indicated by both the Funnel plot and Egger’s test. These findings suggest that the results for these specific subcomponents are more robust and less influenced by potential biases in the published literature. However, for inhibition, the number of included studies was too small to conduct a reliable analysis of publication bias, making it impossible to evaluate bias for this subcomponent.

We acknowledge the limitation of this incomplete assessment of publication bias, particularly in the case of inhibition, where the small number of studies limits the ability to draw firm conclusions about bias across all study domains.

We have added this information in the Discussion section (“Third, although publication bias was reduced through sensitivity analysis, further studies with more rigorous methodologies are needed to validate these findings.”)

Discussion

The authors repeatedly mentioned a lack of significant correlations between VR-based assessments and traditional tests, particularly for attention and inhibition, but they did not provide a thorough explanation of these findings and the reasons behind them. For example,

#12. The risk of bias, particularly in patient selection, is described but not discussed in relation to how it might impact findings on attention, inhibition, and cognitive flexibility.

Response #12: We appreciate the reviewer’s concern regarding the risk of bias in patient selection and how it might impact findings related to attention, inhibition, and cognitive flexibility.

As part of our analysis, we conducted a sensitivity analysis by excluding low-quality studies identified through the risk of bias assessment. This sensitivity analysis demonstrated that statistically significant correlations were maintained across the full dataset, even after excluding these lower-quality studies. This suggests that the low-quality studies did not unduly influence the overall results.

However, we were unable to perform sensitivity analyses for the subcomponents of executive function (i.e., attention, inhibition, and cognitive flexibility) after excluding the low-quality studies. The exclusion of these studies reduced the sample sizes for each subcomponent to a level where meaningful analysis was no longer possible. To maintain sufficient statistical power, we decided not to exclude studies based on quality when analyzing the subcomponents individually.

In light of these limitations, we acknowledge that the risk of bias in patient selection may still have an impact on the results for individual subcomponents, particularly in the domains of attention, inhibition, and cognitive flexibility.

We have added this information in the Discussion section (“First, the limited number of studies prevented sensitivity analyses for all subcomponents of executive function, which reduced the robustness of the findings.”)

#13. The authors described high heterogeneity but did not adequately discuss what factors contributed to this heterogeneity.

Response #13: We appreciate the reviewer’s feedback regarding the discussion of heterogeneity. In response, we have expanded our explanation of the potential sources of heterogeneity in the revised manuscript.

We conducted several subgroup analyses to investigate factors that might contribute to the observed high heterogeneity across the included studies. First, we analyzed whether age (children vs. adults) moderated the correlation between VR-based and traditional assessments. The results indicated that age did not significantly moderate the outcomes, suggesting that age differences were not a major factor in the variability observed.

Next, we examined the clinical status of the sample (categorized as combined, clinical, or healthy). While clinical status did not moderate the results for overall executive function or cognitive flexibility, it was a significant moderator for attention. Specifically, clinical and combined populations exhibited different patterns of correlation compared to healthy populations.

We also explored whether the type of VR-based assessment (adaptations of traditional tools vs. newly developed tasks) moderated the results. However, this factor did not significantly influence the correlations in any of the analyses.

Despite these efforts, the sources of heterogeneity were not fully explained, which we acknowledge as a limitation.

We have added this information in the Discussion section (“Lastly, the study observed high heterogeneity, indicating variability in the results. Future research should aim to investigate additional factors contributing to this variability and focus on enhancing the reliability of VR-based assessments across diverse populations and contexts.”)

#14. Although the discussion briefly discussed the potential benefits of VR, it still lacks of discussion on the specific advantages relating to real-world applications.

Response #14: Thank you for your insightful comment regarding the need for a more detailed discussion of VR’s specific advantages in real-world applications. In the revised manuscript, we have expanded the discussion to emphasize the practical benefits of VR in real-world settings.

One key advantage of VR-based assessments is their enhanced ecological validity. Unlike traditional paper-and-pencil tests, which often fail to replicate real-world tasks, VR environments immerse individuals in scenarios closely resembling everyday activities, such as navigating a virtual home or completing practical tasks. This allows for more realistic and relevant assessments of executive functions, including planning, memory, and cognitive flexibility, improving the prediction of real-world behaviors.

Additionally, VR offers a high degree of customizability and adaptability. VR scenarios can be tailored to meet the specific needs of different populations, such as adjusting tasks for cognitive abilities, age groups, or clinical conditions like ADHD. This adaptability ensures that the evaluation reflects the unique challenges individuals face in their daily lives, enhancing the relevance and applicability of the assessment results.

These real-world applications demonstrate the significant potential of VR technology in both clinical and research settings. By providing a more practical, accurate, and immersive method of assessment, VR-based tools offer a valuable alternative to traditional methods, better suited to capturing the complexities of executive function in real-world contexts.

We have added this information in the Discussion section (“In fact, VR assessments offer immersive, interactive environments that simulate real-world tasks (Parsons 2015), providing more realistic and relevant measures of executive functions. Furthermore, VR offers customizability, allowing assessments to be tailored to the specific needs of diverse populations (Jansari et al. 2014). This ensures that evaluations reflect real-world challenges and cognitive demands. These findings demonstrate the potential of VR technology in clinical and research contexts, offering a practical and accurate alternative to traditional methods.”)

#15. The authors discussed VR adaptations of traditional tests but failed to critically assess whether these adaptations might have influenced the results.

Response #15: In response, we have now addressed the potential influence of VR adaptations on the results. As noted in previous studies, there are two types of VR-based assessments: direct adaptations of traditional paper-and-pencil tests and novel VR assessments that differ from the original format. In cases where traditional assessments were adapted into VR, the use of the VR system itself did not significantly impact the results, leading to high correlations with paper-based tests. This suggests that once users are accustomed to the VR environment, the system itself does not interfere with cognitive function evaluations, allowing for comparable results across both formats. We have added this information in the Discussion section (“Previous studies have shown that some VR-based assessments were direct adaptations of paper-and-pencil tests, while others employed entirely new virtual scenarios to assess executive function. In the former case, the use of VR equipment did not significantly affect the assessment outcomes, resulting in strong correlations with traditional neuropsycho-logical evaluations. This suggests that once individuals adapt to the VR system, the new medium does not substantially influence cognitive function assessments (Parsons and Courtney 2014; Voinescu et al. 2023).”)

#16. Although the authors mentioned the need for further refinement of VR assessments, they did not provide clear recommendations for future research.

Response #16: In response to your suggestion, we have refined our recommendations for future research by emphasizing the need to consider both traditional cognitive measures, such as task scores and reaction times, alongside other diverse outcomes that VR systems can provide. For instance, VR can track motor-related metrics like arm movements and speed during tasks, which, based on embodied cognition theory, are closely linked to cognitive function. These additional metrics could complement traditional cognitive assessments, and future VR systems should aim to incorporate and present a variety of such indicators to offer a more comprehensive evaluation of cognitive abilities.

We have added this information in the Discussion section (“Moreover, VR can capture additional metrics beyond traditional measures of cognitive function, such as task scores or reaction times. For example, it can track arm movements and speed when using controllers, which have been reported to be closely related to cognitive function from the perspective of embodied cognition theory. These motor-related metrics can play a supportive role in assessing cognitive functions (Ribeiro et al. 2024), suggesting that future VR systems should incorporate a variety of indicators to provide a more comprehensive assessment of cognitive abilities.”)

#17. Moreover, they did not discuss well about the implications of the study.

Response #17: In response, we have now addressed the implications of the study more thoroughly in the revised manuscript. Specifically, we discussed the potential influence of VR adaptations, noting that when traditional paper-and-pencil tests were directly adapted into VR, the system did not significantly impact the results, leading to strong correlations with traditional assessments. This suggests that once users are accustomed to the VR environment, the system itself does not interfere with the evaluation of cognitive functions. Furthermore, we expanded on the implications for future research by emphasizing the need for VR assessments to consider not only traditional cognitive measures, such as task scores and reaction times, but also other diverse outcomes that VR systems can capture, like motor-related metrics (e.g., arm movements and speed), which are closely linked to cognitive function according to embodied cognition theory. Incorporating these additional metrics could enhance the comprehensiveness of VR-based cognitive assessments.

Round 2

Reviewer 1 Report

Comments and Suggestions for Authors

I would like to express my appreciation for considering my comments and suggestions during the revision of your paper. The work of revision has been well done. I believe the manuscript is suitable for publication.

Author Response

#1. I would like to express my appreciation for considering my comments and suggestions during the revision of your paper. The work of revision has been well done. I believe the manuscript is suitable for publication.

Response#1: Thank you for your supportive comment. 

Reviewer 2 Report

Comments and Suggestions for Authors

Thank you very much for the authors’ efforts in addressing my comments. This paper is quite close to being ready for publication. However, I still have an important comment regarding the Screening and Data Extraction Process. The authors described, “In cases where discrepancies arose during the screening process, the two reviewers discussed the differences. If they were unable to reach a consensus, a third reviewer was consulted to resolve the disagreement.”

However, no information on inter-rater reliability (IRR) for this process or the data extraction phase has been provided. Reporting IRR values (e.g., Cohen’s kappa) would enhance the transparency of the screening and data extraction process, as it reflects the agreement between reviewers.

Comments on the Quality of English Language

Minor editing of English language required.

Author Response

#1. Thank you very much for the authors’ efforts in addressing my comments. This paper is quite close to being ready for publication. However, I still have an important comment regarding the Screening and Data Extraction Process. The authors described, “In cases where discrepancies arose during the screening process, the two reviewers discussed the differences. If they were unable to reach a consensus, a third reviewer was consulted to resolve the disagreement.”

However, no information on inter-rater reliability (IRR) for this process or the data extraction phase has been provided. Reporting IRR values (e.g., Cohen’s kappa) would enhance the transparency of the screening and data extraction process, as it reflects the agreement between reviewers.

Response#1: Thank you for raising your concern. We have added the IRR values to the Result section.

#2. Minor editing of English language required.

Response#2: Thank you for your comment. We have reviewed and revised the manuscript for clarity and correctness in English.

Round 3

Reviewer 2 Report

Comments and Suggestions for Authors

 Accept in present form.